# The implicit power of positive thinking: The effect of positive episodic simulation on implicit future expectancies

Rachel J. Anderson [ID]*, J. Helgi Clayton McClure [ID]¤a, Emma Bishop, David Howe¤b, Kevin J. Riggs, Stephen A. Dewhurst

School of Psychology and Social Work, University of Hull, Hull, United Kingdom

¤a Current address: School of Education, Language and Psychology, York St John University, York, United Kingdom
¤b Current address: Leeds Beckett University, Leeds, United Kingdom
* rachel.anderson@hull.ac.uk

## Abstract

Previous research demonstrating that positive episodic simulation enhances future expectancies has relied on explicit expectancy measures. The current study investigated the effects of episodic simulation on implicit expectancies. Using the Future Thinking Implicit Relational Assessment Procedure (FT-IRAP), participants made true/false decisions to indicate whether or not they expected positive/negative outcomes after adopting orientations consistent or inconsistent with an optimistic disposition. The outcome measure, $D_{IRAP}$, was based on response time differences between consistent and inconsistent blocks. Participants then engaged in either positive simulation training, in which they imagined positive future events, or a neutral visualisation task before repeating the FT-IRAP twice following 10-minute intervals. Positive simulation training increased $D_{IRAP}$ scores for *don't-expect-negative* trials–boosting participants' readiness to affirm that negative events were unlikely to happen to them. Although findings did not generalise across all trial types, they show potential for positive simulation training to enhance implicit future expectancies.

## Introduction

Episodic simulation refers to the capacity to construct detailed mental representations of possible specific events within one's personal future [1]. Episodic simulation is posited to serve an adaptive purpose whereby it forms the foundation for other facets of prospection, such as the expectancies one holds about whether future events are likely to occur and whether one will achieve one's personal goals [2]. Arguably, the usefulness of episodic simulations depends on the ease with which they come to mind and the extent to which they contain vivid episodic details [3,4]; in which case, training to promote effective episodic simulation could serve as a mechanism by which other aspects of prospection, such as future expectancies and intentions, can be modified. Research has shown that future expectancies can be enhanced by repeatedly engaging in episodic simulation about potential positive future events [e.g. 5,6]. A limitation of

**Funding:** This work was supported by the UKRI Economic and Social Research Council, grant number ES/R007152/1 awarded to RJA, SAD and KJR. The funders had no role in study design, data collection and analysis, decision to publish, or preparation of the manuscript.

**Competing interests:** The authors have declared that no competing interests exist.

previous research, however, is that it has focused on explicit measures of future expectancies in the form of self-reports. To overcome this limitation, the current study investigated the effects of a brief induction of positive episodic simulation on implicit expectancies about the future.

Developing methods to modify prospective thinking is important because difficulties in prospection have been linked to the experience of psychological distress, including depression. Individuals evidencing both clinical and non-clinical levels of depressive symptomatology exhibit difficulties in episodic simulation tasks, characterised by a shift away from the generation and use of episodic details in favour of semantic information [7–10]. Moreover, it seems that positive episodic simulation is particularly difficult for depressed individuals; that is, they have difficulty generating vivid simulations of positive, but not negative, future occurrences [11–17]. In addition to difficulties with simulation, depression is associated with biased future expectancies. Negative future events are believed to be more likely to occur, whilst expectations about positive future events, including achievement of personal goals, are muted [13,15,18–20]. Expectations about how positive future events will make one feel are also biased; both anticipated (how one expects to feel if the future event were to happen) and anticipatory (the in-the-moment feelings one experiences when thinking about the potential future experience) pleasure are blunted when thinking about potential positive events and goal achievement [19,21–25]. Roepke and Seligman [26] argued that these biased prospections might play a causal role in depression, whereby if one has difficulty envisaging a positive future and expects that things will continue to turn out negatively then sadness and hopelessness are an unsurprising response. Furthermore, these biased simulations and expectations may underpin the motivational difficulties evident in depression; an individual is unlikely to feel motivated to work towards an event/goal that they struggle to envisage, foresee as unlikely to occur and for which they struggle to anticipate the pleasure its occurrence would bring.

Converging theory and evidence suggests that an individual's ability to simulate vivid and episodically detailed future events may underpin other facets of prospection, such as the expectancies one holds about the future, and that these interrelated prospection biases may have a causal role in the development and/or maintenance of depression. Thus, it seems logical that training to promote effective episodic simulation of positive future events could serve as a fruitful avenue for therapeutic intervention. Studies, using both samples of non-depressed individuals and those experiencing elevated depressive symptomatology, provide evidence for the effects of training in episodic simulation as a method of improving both the episodic detail contained within simulations themselves and, in turn, one's expectancies about the future. Brief inductions of episodic simulation have been shown to increase the episodic detail and vividness of future events [5,27], make future episodes feel more plausible, controllable and likely to occur [5,6,28–30], and increase the anticipated and anticipatory pleasure associated with future events [6]. Similar effects have been found with Future Specificity Training, a structured intervention designed to improve episodic simulation and its associated detail and mental imagery [31,32]. The effects of Future Specificity Training on episodic simulation were maintained at a 2-week follow-up [31].

The growing body of literature outlined thus far supports the notion that training in positive episodic simulation has the potential to improve one's expectancies about possible future events. A number of these studies demonstrated that engaging in episodic simulation modified the future expectancies for those same events [e.g. 6,30]. However, there is also evidence to suggest that the effects of positive episodic simulation can generalise to expectancies about future events beyond those that form part of the simulation training itself. In other words, the individuals' optimistic orientation is modified. For instance, Boland et al [5] used two versions of their positive simulation training task; one was designed to promote simulations that were conceptually related to the events about which expectancy judgements were provided, whilst

the other was designed to promote unrelated simulations. Both tasks led to similar changes in expectancies about potential future events. Furthermore, Hallford, Yeow et al [32] found that Future Specificity Training led to some improvements in more generalised expectations about one's conceptual future self, such as feeling successful or being physically fit, in addition to improvements in participants' ability to engage in episodic simulation.

To date, the literature examining how episodic simulation can modify one's expectancies about potential future events is limited by its reliance on self-report measures of future expectancies. Participants explicitly rate their expectancies about a future episode or state using Likert-type scales. Whilst this explicit style of questioning is easy to administer, it is open to the possibility of response bias or demand characteristics. In order to overcome this limitation, the current study investigated the effect of positive episodic simulation on an implicit measure of future expectancies. Implicit measures are advantageous over explicit measures because they reduce the likelihood of demand characteristics and also target automatic beliefs [33,34]. For this purpose, we used an implicit measure of future expectancies created by Kosnes et al [35].

Kosnes et al.'s [35] task was adapted from the Implicit Relational Assessment Procedure [IRAP: 36] and termed the Future Thinking IRAP (FT-IRAP). The task requires participants to provide responses that are consistent or inconsistent, in alternating blocks, with an optimistic orientation. In each block, participants respond 'true' or 'false' on trials presenting either 'I expect' or 'I don't expect' in combination with one of six positive future experiences (e.g. friendship, enjoyment) and six negative future experiences (e.g. worry, loneliness). In a consistent block, the participant has to respond 'true' to items like 'I expect success' and 'I don't expect failure', and 'false' otherwise. In an inconsistent block they have to respond in the opposite manner (i.e., reflecting a *pessimistic* orientation). The rationale of the FT-IRAP is that response times are faster when the orientation is consistent with the valence of the future experience. For example, participants are faster at confirming that they expect a positive experience if they adopt an optimistic orientation rather than a pessimistic orientation. An outcome measure, $D_{IRAP}$, is then computed based on the response time difference between consistent and inconsistent blocks for each of the four trial types (i.e., expect-positive, expect-negative, don't-expect-positive, don't-expect-negative). Rather than simply measuring the strength of the association between orientation and the imagined experiences, $D_{IRAP}$ measures the *direction* of the association. For example, high scores are indicative of an optimistic orientation. Although participants are instructed to adopt either an optimistic or a pessimistic orientation, the fact that the outcome measure is based on differences in response times between consistent and inconsistent blocks across four different trial types makes it unlikely that results will be subject to demand characteristics [see 37 for discussion of how demand characteristics are reduced by implicit tasks and response time measures].

Using the task developed by Kosnes et al. [35], the current study investigated the impact of a brief positive simulation training (PST) induction on implicit expectancies about the future. Participants completed the FT-IRAP before completing one of two tasks, PST or a neutral imagery task. This allowed us to test whether it is episodic simulation, rather than generic imagery engagement per se, that impacts future expectancies. The use of a neutral imagery condition also enabled us to investigate whether PST enhances implicit expectancies relative to a baseline level. Subsequently, participants completed the FT-IRAP a further two times, once immediately after the PST/neutral imagery task and again after completion of a 10-minute filler task. This allowed us to establish whether any beneficial effects of PST were transitory or whether they remained after a short delay. In line with theories of prospection that suggest episodic simulation underpins the expectancies one holds about the future, it was hypothesised that PST would lead to an increase in FT-IRAP scores (indexing enhanced optimistic

orientation) relative to the neutral imagery task. A secondary aim of the current study was to investigate whether any effects of PST differed as a function of depressive symptomatology; for this purpose, we treated severity of depressive symptoms as a continuous variable to reflect the evidence that they exist along a continuum of severity throughout the population [38]. If positive episodic simulation is to form a useful tool for therapeutic intervention, it is important to establish that it is effective in individuals with higher levels of depressive symptomology.

## Materials and method

### Participants

Ethical approval was granted by the University of Hull Faculty of Health Sciences Ethics Committee (Ref FHS216) and all participants provided written informed consent ahead of participation. A target sample size of 70–80 was determined based on prior studies using comparable versions of the IRAP procedure with two participant groups [35,39]. 82 students at the University of Hull participated in the study, between 29[th] September 2022 and 2[nd] February 2023, in return for either course credit or a £12 Amazon e-voucher. Nine participants were excluded based on standard IRAP/FT-IRAP criteria [40], leaving 73 participants whose data were included in analyses. Those in the final sample ranged from 18 to 50 years old ($M$ = 22.8, $SD$ = 7.6). Depressive symptoms were measured using the Center for Epidemiologic Studies Depression Scale–Revised [41], with scores ranging from 0 to 54 ($M$ = 19.9, $SD$ = 13.0).

Participants were randomly assigned to one of two experimental groups. The PST group comprised 36 participants, of which 28 identified as female and 8 as male, ranging in age from 18 to 49 ($M$ = 23.0, $SD$ = 7.7). The neutral imagery group comprised 37 participants, of which 31 identified as female and 6 as male, ranging in age from 18 to 50 ($M$ = 22.6, $SD$ = 7.5). The two groups did not differ significantly in terms of age, $t$ (71) = 0.86, $p$ = .39, or gender, $\chi^2$ (1,71) = .42, $p$ = .51. All participants completed the CESD-R [41] to assess severity of depressive symptoms. They were also asked to indicate whether they were currently receiving treatment for depression, had previously received treatment, or had never received treatment. These data are shown in Table 1. The groups did not differ significantly in terms of mean CESD-R score, $t$ (71) = .09, $p$ = .93, the number of participants meeting criteria for non-clinical depression levels of (dysphoria), $\chi^2$ (1,71) = .01, $p$ = .90, or treatment status, $\chi^2$ (2,71) = .26, $p$ = .88.

### Materials

**Center for Epidemiologic Studies Depression Scale–Revised (CESD-R).** The CESD-R (41) is a 20-item inventory which assesses depressive symptoms in nine different clusters as defined by the Diagnostic and Statistical Manual of Mental Disorders [DSM-5; 42]. Items are scored on a five-point scale representing the frequency that an individual has experienced that symptom over the previous 1–2 weeks (0 = 'Not at all or less than 1 day' to 4 = 'Nearly every day for 2 weeks'). Responses are summed to produce a total score from 0–80, with higher

**Table 1. CESD-R data and treatment status as a function of experimental group.**

|  | PST ($n$ = 36) | Neutral imagery ($n$ = 37) |
|---|---|---|
| Mean CESD-R score | 20.06 ($SD$ = 12.87) | 19.78 ($SD$ = 13.27) |
| Meeting criteria for dysphoria | 17 | 18 |
| Currently receiving treatment | 2 | 3 |
| Previously received treatment | 9 | 8 |
| Never received treatment | 25 | 26 |

values indicating elevated depressive symptomatology. The CESD-R has demonstrated strong psychometric properties in community samples [43].

**Future Thinking Implicit Relational Assessment Procedure (FT-IRAP).** The stimuli and procedure used for the FT-IRAP task were based on those used by Kosnes et al. [35]. Participants were presented with 'sample' stimuli at the top of the screen and 'target' stimuli in the centre of the screen. Each trial consisted of one sample stimulus, either 'I expect' or 'I don't expect', along with one target stimulus which was either a positive future outcome (e.g., 'happiness') or a negative future outcome (e.g., 'sadness'). The basic structure of the task is outlined in Fig 1. There were 12 target stimuli in total, of which 6 were positive (friendship, enjoyment, happiness, wealth, success, and love) and 6 were negative (worry, loneliness, failure, stress, sadness and illness). On every trial, participants had to respond either 'true' (Z key) or 'false' (M key). Before each block of trials, participants were instructed to respond either as though they expect positive events to occur and do not expect negative events to occur ('consistent' blocks) or to respond as though they do not expect positive events to occur and do expect negative events to occur ('inconsistent' blocks). Therefore, consistent blocks required the following pattern of responses: I expect–positive–*true*, I expect–negative–*false*, I don't expect–positive–

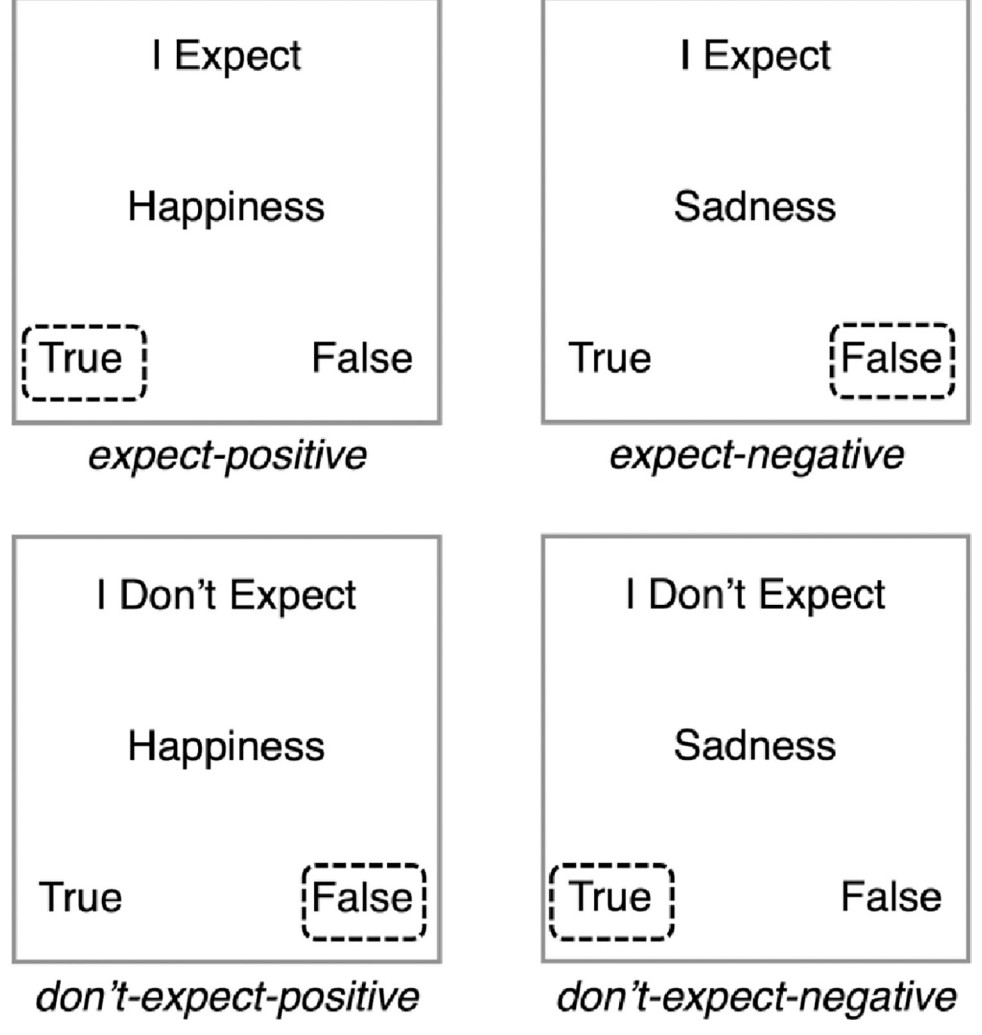

**Fig 1. Structure of FT-IRAP task, showing four trial types and correct consistent-block responses.**

*false*, I don't expect–negative–*true* (illustrated in Fig 1). The converse pattern of responses was required in inconsistent blocks.

Each FT-IRAP procedure involved four blocks of trials, alternating between consistent and inconsistent instructions. Block order was counterbalanced so that half of participants completed a consistent block first and the other half an inconsistent block first. Each block consisted of 24 trials: 6 each of the four possible pairings of sample and target stimuli (i.e., the four trial types: expect-positive, expect-negative, don't-expect-positive, don't-expect-negative; Fig 1). Therefore, each target stimulus was presented twice per block (once with 'I expect', once with 'I don't expect'). Trials were presented in a randomised order. For each trial, sample stimuli, target stimuli and response options remained onscreen until participants made a response. If participants made an incorrect response, they were presented with a red cross in the centre of the screen, which remained until the correct response was made. A correct response cleared the screen for 400ms before the next trial began. Response time and accuracy were recorded for each trial. At the end of each block, participants were given feedback on their average response time and accuracy for that block, with a reminder that they should aim for < 2 seconds average response time and at least 80% accuracy.

Response time data from each FT-IRAP were used to compute '$D_{IRAP}$' scores for each participant based on the method outlined by Barnes-Holmes et al [40]. The following steps were carried out using a bespoke algorithm compiled in R. First, all practice trials and trials with response times >10,000 ms were removed. A participant's entire data set was excluded if >10% of their trials had response times of 300ms or less ($n = 1$) or if they had registered less than 80% accuracy for any individual test block ($n = 8$).

After applying these exclusion criteria, mean response times were calculated for each of the four trial types (expect-positive, expect-negative, don't-expect-positive, don't-expect-negative) within each task block (i.e., 16 initial mean values per participant, per time point). Eight difference scores were then computed–one for each trial type and pair of test blocks–by subtracting the mean consistent-block RT from the mean inconsistent-block RT (e.g., mean RT for 'I expect–positive' trials in the first inconsistent block minus mean RT for 'I expect–positive' trials in the first consistent block). Each difference score was then divided by the corresponding pooled standard deviation of raw RTs in the relevant two blocks. These were then averaged across repeated block pairs, producing four overall $D_{IRAP}$ scores (one for each trial type) at each time point. The resulting $D_{IRAP}$ scores could vary between –2 and +2, with positive values indicating an optimistic orientation and negative values a pessimistic orientation. These served as our primary dependent measure.

**Jigsaw task.** Participants were invited to choose one of 12 famous world landmark scenes (e.g., the Eiffel Tower, Niagara Falls) on an interactive jigsaw app for iPad [44]. Each jigsaw comprised 252 pieces. Participants had to move the pieces into place with their finger from the bottom of the screen and were given 10 minutes to complete as much of the jigsaw as possible.

**Visual Analogue Mood Scales (VAMS).** Positive and negative affect were measured using printed visual analogue scales, where participants had to mark on a 100mm line how they felt in that moment [45]. The positive scale ranged from 'Totally not in a positive mood' (0) to 'Very positive mood' (100); the negative scale ranged from 'Totally not dejected, "down", sad, depressed' (0) to 'Very dejected, "down", sad, depressed' (100).

**Positive Simulation Training (PST).** This task, adapted from that used by Boland et al [5], required participants to simulate a series of positive future events in response to cue words provided onscreen (e.g., 'family', 'proud'). Each cue word appeared onscreen for 15 seconds, during which participants were instructed to imagine a positive future event relating to that word in as much detail as possible. Participants performed five practice trials before starting the main experimental block, which consisted of 30 trials (10 cue words repeated three times

in a randomised order). At the end of the practice and experimental blocks, participants were asked to rate, on average, how vividly they were able to imagine the events they had just simulated and how emotional the events were (0 = 'Not at all vivid/emotional', 6 = 'Extremely vivid/emotional').

**Neutral imagery task.** This task required participants to visualise neutral scenes in response to descriptions provided onscreen (e.g., 'the baggage claim area at an airport'). The task featured 10 items taken from Nolen-Hoeksema and Morrow [46], presented three times each in a randomised order. Stimulus duration was 15 seconds and participants completed five practice trials before starting the main experimental block of 30 trials. At the end of each block, participants were asked to rate how vividly they were able to imagine the neutral scenes and how emotional they were using the same scales as employed in the PST task.

## Procedure

Participants completed all tasks individually, with the researcher providing instructions prior to each task and remaining present throughout. The CESD-R and VAMS were completed on paper. The FT-IRAP and PST/neutral imagery tasks were run in OpenSesame [47].

After providing informed consent, participants completed the CESD-R before being introduced to the FT-IRAP task with completion of two practice blocks under the researcher's supervision. They then completed the first four-block FT-IRAP (Time 1), taking self-paced rest breaks between blocks as required. Participants were then instructed to move on to the jigsaw task, with the researcher setting a timer for 10 minutes. After this, the first VAMS was administered, followed by either the PST or neutral imagery task. Participants then completed a second VAMS, followed by the second FT-IRAP (Time 2), and then continued with the jigsaw task for a further 10 minutes. After this they completed a third VAMS, followed by a third and final FT-IRAP (Time 3), and finally, a fourth VAMS to conclude the experiment.

## Results

Descriptive statistics are presented in Table 2 ($D_{IRAP}$ scores) and Table 3 (VAMS, PST/neutral imagery task ratings). All post-hoc tests reported below (i.e., comparing groups separately for each trial type) have been Bonferroni-corrected (i.e., α = .0125), with *p*-values adjusted accordingly. In order to investigate whether any effects of PST differed as a function of depressive symptomatology, CESD-R scores were entered as covariates in all group comparisons.

### Baseline $D_{IRAP}$ scores

First, we evaluated $D_{IRAP}$ scores at Time 1 to check for the presence of existing optimistic bias across experimental groups. We computed a 4 (trial type: expect-positive vs. expect-negative vs. don't-expect-positive vs. don't expect negative) × 2 (group: PST vs. neutral imagery) mixed

**Table 2. Mean (SD) $D_{IRAP}$ values by group, time and trial type.**

| Trial Type: | Time 1 | | | | Time 2 | | | | Time 3 | | | |
|---|---|---|---|---|---|---|---|---|---|---|---|---|
| | *+Pos* | *+Neg* | *-Pos* | *-Neg* | *+Pos* | *+Neg* | *-Pos* | *-Neg* | *+Pos* | *+Neg* | *-Pos* | *-Neg* |
| PST | *0.65* | *-0.06* | *0.15* | *0.05* | *0.72* | *-0.22* | *0.09* | *0.26* | *0.51* | *-0.19* | *0.09* | *0.21* |
| | *(0.47)* | *(0.48)* | *(0.58)* | *(0.44)* | *(0.37)* | *(0.48)* | *(0.43)* | *(0.49)* | *(0.43)* | *(0.51)* | *(0.49)* | *(0.43)* |
| Neutral Imagery | 0.51 | -0.25 | 0.11 | 0.27 | 0.44 | -0.24 | 0.19 | 0.12 | 0.44 | -0.27 | 0.05 | 0.15 |
| | (0.37) | (0.43) | (0.39) | (0.40) | (0.39) | (0.37) | (0.51) | (0.41) | (0.41) | (0.44) | (0.47) | (0.40) |

*Note*. +Pos = expect-positive; +Neg = expect-negative; -Pos = don't-expect-positive; -Neg = don't-expect-negative.

**Table 3. Mean (SD) Mood Ratings (VAMS) and PST / imagery task ratings.**

| | VAMS 1 | | VAMS 2 | | VAMS 3 | | VAMS 4 | | PST/Imagery Task Rating | |
|---|---|---|---|---|---|---|---|---|---|---|
| | Pos | Neg | Pos | Neg | Pos | Neg | Pos | Neg | Vividness | Emotionality |
| PST | 66.97 (15.45) | 21.14 (18.24) | 69.83 (21.67) | 19.25 (21.47) | 70.72 (21.67) | 17.69 (17.56) | 76.53 (16.57) | 16.31 (17.55) | 4.39 (1.05) | 4.00 (1.64) |
| Neutral Imagery | 67.95 (14.75) | 21.19 (16.33) | 65.95 (16.66) | 22.73 (17.78) | 73.84 (16.36) | 16.78 (14.88) | 73.08 (18.71) | 17.00 (17.10) | 4.44 (1.46) | 1.20 (1.27) |

*Note.* VAMS = Visual Analogue Mood Scale scores; Pos = positive [0–100]; Neg = negative [0–100].

ANOVA with repeated measures on the trial type independent variable. This analysis found a significant main effect of trial type ($F_{(3, 213)} = 36.2$, $p < .001$, $\eta_p^2 = .34$) and significant Trial Type × Group interaction ($F_{(3, 213)} = 3.23$, $p = .023$, $\eta_p^2 = .04$) but no significant main effect of group ($F < 1$, $p > .50$).

Marginal means and 95% confidence intervals for these data are shown in Fig 2. These reveal existing optimistic bias on some, but not all, trial types: On expect-positive trials, responses were optimistic (i.e., $D_{IRAP} > 0$) in both the PST group ($M = .65$, 95% CI [.51, .79]) and neutral imagery group ($M = .51$, 95% CI [.38, .65], $t_{(71)} = 1.35$, $p_{bonferroni} = .72$). On expect-negative trials, responses tended to be negative (i.e., pessimistic) in the PST group ($M = -.06$, 95% CI [-.21, .09]) and were negative in the neutral imagery group ($M = -.25$, 95% CI [-.40, -.10], $t_{(71)} = 1.81$, $p_{bonferroni} = .30$). On don't-expect-positive trials, there was no clear evidence of bias for either the PST ($M = .15$, 95% CI [-.02, .31]) or neutral imagery group ($M = .11$, 95% CI [-.05, .27], $t_{(71)} = 0.35$, $p_{bonferroni} = 1.00$). Finally, on don't-expect-negative trials, the PST group showed no clear bias ($M = .05$, 95% CI [-.09, .19]) while the neutral imagery group

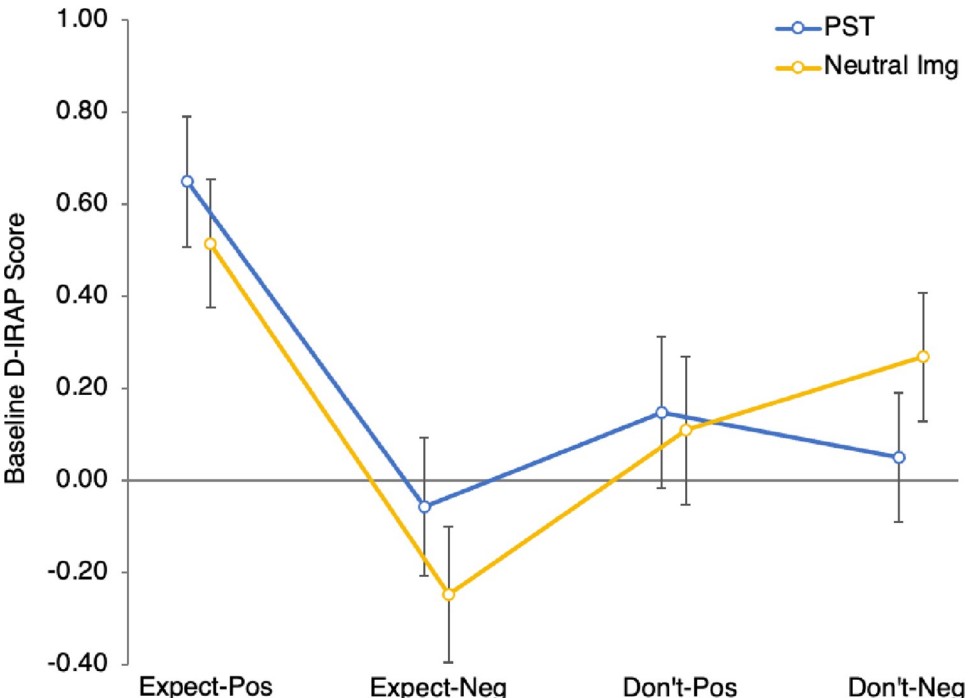

**Fig 2. $D_{IRAP}$ scores at Time 1 by trial type and group (means with 95% CIs).**

showed an optimistic bias ($M$ = .27, 95% CI [.13, .41], $t_{(71)}$ = 0.35, $p$ = .031, $p_{\text{bonferroni}}$ = .12). To correct for these disparities, subsequent analyses were performed on difference scores relative to baseline, rather than raw $D_{\text{IRAP}}$ data.

## Immediate effect of PST

We next ascertained whether engaging in PST, relative to the neutral imagery task, improved implicit expectancies about future events when assessed immediately after the manipulation. Difference scores were computed by subtracting Time 1 $D_{\text{IRAP}}$ values from Time 2 $D_{\text{IRAP}}$ values. We then conducted a 4 (trial type) × 2 (group) mixed ANCOVA on the change scores, with square root transformed CESD-R score as a covariate (raw CESD-R skewness = .73, SE = .28, $Z > 2$, kurtosis = -.34, SE = .56, $Z < 2$; transformed skewness = .03, kurtosis = -.40, both $Z$s < 2).

In this analysis, main effects were non-significant for both trial type ($F_{(3, 207)}$ = 1.36, $p$ = .26) and group ($F < 1$, $p > .50$), but a significant Trial Type × Group interaction emerged ($F_{(3, 207)}$ = 3.93, $p$ = .009, $\eta_p^2$ = .06), as illustrated in Fig 3. Post-hoc tests confirmed that scores increased for PST participants relative to controls for *don't-expect-negative* trials ($t_{(69)}$ = 2.71, $p_{\text{bonferroni}}$ = .032, $d$ = .68), but not for the other three trial types ($ts < 1.5$, $p_{\text{bonferroni}} > .50$). Neither the main effect of CESD-R score ($F < 1$, $p > .50$), nor its interaction with trial type ($F_{(3, 207)}$ = 2.23, $p$ = .09), were significant.

## Maintenance of PST effect over a short delay

To establish whether the identified effect of PST (on *don't-expect-negative* trials) was maintained after the 10-minute filler task, we computed change scores between Time 1 (baseline)

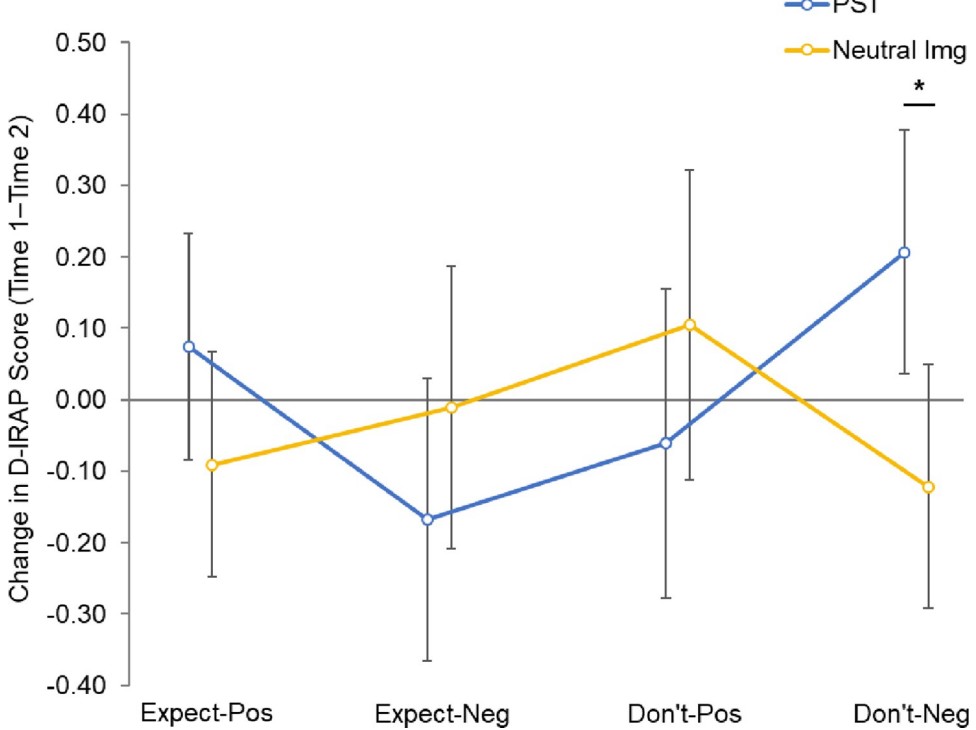

**Fig 3. $D_{\text{IRAP}}$ change scores (Time 1–Time 2) by trial type and group (means with 95% CIs).**

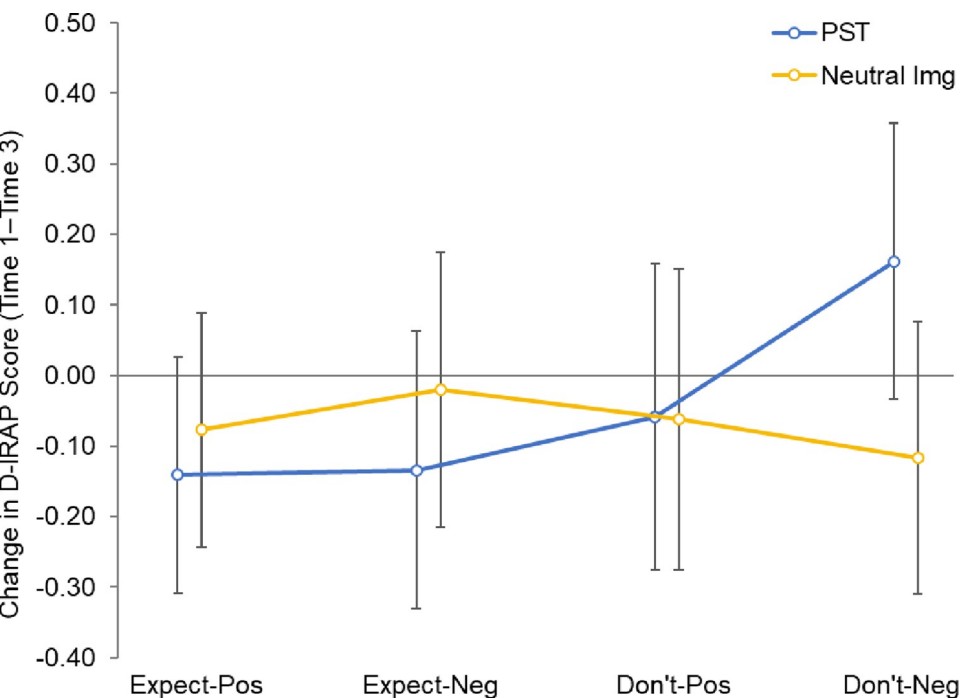

**Fig 4. $D_{IRAP}$ change scores (Time 1–Time 3) by trial type and group (means with 95% CIs).**

and Time 3 (end of experiment) and conducted a similar 4 (trial type) × 2 (group) mixed ANCOVA with transformed CESD-R score as a covariate.

This analysis again found non-significant main effects for both trial type ($F_{(3, 210)}$ = 1.40, $p$ = .24) and group ($F < 1$, $p > .50$), and a non-significant Trial Type × Group interaction ($F_{(3, 210)}$ = 1.78, $p$ = .15). Marginal means and 95% confidence intervals are shown in Fig 4. Again, neither the main effect of CESD-R score, nor its interaction with trial type, were significant ($F$s $< 1.3$, $p$s $> .25$).

## Changes in mood

We also analysed VAMS scores relative to baseline to identify any fluctuations in positive and/ or negative affect which might have influenced FT-IRAP results. First, we ran a 2 (VAMS valence: positive vs. negative) × 2 (group: PST vs. neutral imagery) mixed ANCOVA, with repeated measures on the VAMS valence independent variable and transformed CESD-R score as a covariate, on difference scores derived from VAMS-1 and VAMS-2 (i.e., from immediately before to immediately after the PST / neutral imagery task). If mood substantially altered in this period, apparent PST effects might be attributable to this change. This analysis found no significant main effects ($F$s $< 1$, $p$s $> .70$) and a non-significant VAMS Valence × Group interaction ($F_{(1, 70)}$ = 1.71, $p$ = .20). Neither the main effect of CESD-R score ($F_{(1, 70)}$ = 1.26, $p$ = .27), nor its interaction with VAMS valence ($F < 1$, $p > .60$), were significant. We then repeated this analysis on difference scores between VAMS-1 and VAMS-3 (i.e., from immediately before the PST / neutral imagery task to before the third and final FT-IRAP). Again, there were no significant main effects ($F$s $< 1$, $p$s $> .70$) and a non-significant Valence × Group interaction ($F < 1$, $p > .50$). Neither the main effect of CESD-R score ($F < 1$, $p > .40$), nor its interaction with valence ($F_{(1, 70)}$ = 2.10, $p$ = .16), were significant. It is

therefore unlikely that the observed short-term difference in FT-IRAP scores (for don't-expect-positive trials) can be explained by changes in mood.

## Ratings of vividness and emotionality

To ensure the findings were not confounded by individual differences in PST/neutral imagery task engagement, we compared ratings of vividness and emotionality across the PST and neutral imagery groups. An independent-samples $t$-test on vividness ratings showed no difference ($t_{(70)} = -.19$, $p = .85$), but an equivalent test on emotionality ratings showed a significant difference ($t_{(70)} = 8.13$, $p < .001$, $d = 1.92$): The PST task ($M = 4.00$, 95% CI [3.59, 4.41]) was judged more emotional than the neutral imagery task ($M = 1.65$, 95% CI [1.24, 2.06]). Nonetheless, individuals' emotionality ratings were not correlated with changes in $D_{IRAP}$ scores, either from Time 1 to Time 2 (absolute $r$s $< .21$, $p$s $> .08$) or from Time 2 to Time 3 (absolute $r$s $< .15$, $p$s $> .20$). It is therefore unlikely that the observed short-term difference in FT-IRAP scores (for don't-expect-positive trials) depended on the degree of emotionality experienced.

## Effect of depressive symptom severity

As the secondary aim of the study was to investigate whether effects of PST differed as a function of depressive symptom severity, it was important to determine whether CESD-R score was correlated with any of the other measures. CESD-R score was not significantly correlated with FT-IRAP performance for any time point / trial type ($|r| < .18$, $p > .14$), nor was it correlated with vividness ($r = .19$, $p = .12$) or emotionality ($r = .03$, $p = .82$). However, CESD-R score was positively correlated with negative mood scores at all time points ($r \geq .29$, $p \leq .014$) and negatively correlated with positive mood scores at times 1 and 4 ($r \leq -.25$, $p \leq .033$).

## Discussion

The primary purpose of the current study was to investigate the effect of a brief induction in positive episodic simulation (PST) on an implicit measure of future expectancies, the FT-IRAP. Theoretically, episodic simulation is argued to underpin the expectancies one holds about the future and, thus, training in positive episodic simulation has the potential to improve an individual's optimistic orientation. As such, it was hypothesised that our PST task would lead to an increase in FT-IRAP scores relative to the neutral imagery task. We found that expectancies, indexed by responses for *don't-expect-negative* trials, increased from Time 1 (before induction) to Time 2 (immediately after induction) for participants who completed PST, compared to those who completed the neutral imagery task. However, the change in expectancies across the other three trial types (expect-positive, expect-negative, don't-expect-positive) did not differ significantly between the PST and neutral imagery tasks. Thus, our hypothesis was only partially supported. PST, relative to engaging in a neutral imagery task, enhanced optimistic orientation; however, this effect was only evident for one trial type within the FT-IRAP.

Our findings extend the previous literature, which has demonstrated that training in positive episodic simulation improves one's explicit expectancies about possible future events [5,6,30,32]. A limitation of this existing literature was its reliance on self-report measures of future expectancies, which are open to the possibility of response bias or demand characteristics. Thus, we provide crucial evidence that the effects of PST are not only evident in the overtly stated, explicit, expectancies that one reports about future events but that they do, to some extent, extend to the automatic beliefs one holds about the future. This further supports the theoretical notion that episodic simulation serves an adaptive purpose whereby it forms

the foundation for other facets of prospection, in this case the expectancies one holds about whether future events are likely to occur [2].

Given that the four trial types in the FT-IRAP are assumed to index complementary dimensions of future expectancy, whereby a generally optimistic person should be quick to endorse both 'I expect happiness' and 'I *don't* expect sadness' etc., one might ask why PST-mediated changes were only evident in one of the four cases (don't-expect-negative). It is worth noting here that the present sample showed a variable pattern of $D_{IRAP}$ scores at baseline (see Fig 2). It is possible that the already high $D_{IRAP}$ scores for expect-positive trials (.51–.65) might represent a ceiling effect, leaving little room for improvement. This does not, of course, explain the lack of improvement on expect-negative and don't-expect-positive trials, which were below or indistinguishable from zero at baseline. In these cases, however, the additional cognitive complexity of having to respond 'false' in the consistent (i.e., optimistic) blocks might both have reduced apparent optimistic bias at Time 1 and dampened the effects of PST. Although we followed the procedure developed by Kosnes et al [35], more recent recommendations note the importance of carefully selecting IRAP stimuli to avoid negations that increase cognitive load when responding at speed [48]. In any case, if $D_{IRAP}$ scores across the four trial types reflected the same underlying pattern of future expectancies, one would expect them to be positively intercorrelated, which was not the case in our sample (*r*s < .18, *p*s > .13). These results, then, provide tentative evidence of a PST benefit on implicit future expectancies; yet they also highlight a need to clarify the internal consistency and specificity of IRAP measures when applied in this context.

The findings discussed so far relate to the FT-IRAP scores obtained before (Time 1), and immediately after (Time 2) the experimental manipulation. We also asked participants to complete a further FT-IRAP (Time 3) after a 10-minute filler task; no significant differences were found between FT-IRAP scores across timepoints 1 and 3. This suggests that the increase in implicit expectancies evident as a function of PST is relatively transient. From a therapeutic perspective this is important because PST can only be a useful tool if its effects endure. It is important to note that our study only investigated the relatively short-term endurance of these effects following a single experimental induction. Regular practice of PST is likely to be a feature of any intervention and, arguably, could lead to its effects becoming more sustained. For instance, two studies have investigated the sustained impact of training in episodic simulation on explicit expectancies about the future within the context of a structured intervention [31,32]. These studies found that Future Specificity Training led to increased detail and use of mental imagery within the simulation process and improvements in expectancies about potential positive future events (likelihood of occurrence, control and anticipated/anticipatory pleasure). However, as yet, we do not yet know whether sustained training in episodic simulation impacts implicit expectancies about the future in a similar way. Thus, further work needs to fully delineate the duration of the effects of PST on both implicit and explicit future expectancies; this would provide an empirical basis for therapeutic intervention design with respect to how frequently an individual should practice positive episodic simulation to maintain the therapeutic effects.

A secondary aim of the current study was to investigate whether any effects of PST on implicit expectancies differed as a function of depressive symptomatology. In this regard, we found that the beneficial effect of PST, in comparison to neutral imagery, did not vary as a function of level of depressive symptomatology. This is important because for PST to form a useful tool for therapeutic intervention, its beneficial effects need to be evident across the continuum of depressive experience. Depression represents one, but not the only, form of psychological distress that evidences biases in prospective thinking. These include difficulties in generating vivid and detailed episodic simulations [e.g. 8,9,12,16] and future expectancies that

are pessimistically biased [14,19,22]. These biased prospections are argued to play a causal role in depression [26] and, therefore, finding ways to modify these biases is likely to form a critical component within cognitive-behavioural techniques that target depressive experiences. Thus, our findings add further weight to the argument that training in episodic simulation may be a useful tool in changing the pessimistic outlook evident in individuals experiencing elevated levels of depression.

It is important to acknowledge a number of limitations to the current study. First, we did not include an explicit measure of future expectancies and, as such, we have been unable directly compare the effect of PST on implicit and explicit expectancies within the same participants. Our main reason for this was to avoid making the task overly complex. Future research, therefore, might combine explicit measures with a simplified version of the FT-IRAP that avoids negations. Future research could also include participants who meet the criteria for major depressive disorder. We have argued that the effectiveness of a therapeutic intervention should not vary as a function of depressive symptom level. Extending the investigation to clinically depressed individuals would be consistent with this objective. It is possible that a certain level of symptomology is necessary to avoid baseline ceiling effects in optimistic orientation or to enable participants to experience any benefits of the training. A wider range of depressive symptom levels would allow us to address these issues. A further limitation is that we investigated the effects of a single induction of episodic simulation on implicit expectancies over relatively short intervals. A useful direction for future research would be to investigate the effects of repeated episodic simulation over much longer timeframes and/or to examine the effect of a newly developed intervention, Future Episodic Specificity Training [31,32], on implicit future expectancies.

It is also necessary to acknowledge limitations inherent in the FT-IRAP. For example, the task features multiple presentations of a limited number of stimuli, which may lead to habituation or practice effects, with potential impacts on task engagement. Presenting the same stimuli in different conditions may also result in interference across successive presentations. However, we believe the counterbalancing we employed precluded any systematic effects on task performance. A further limitation is that the FT-IRAP is self-paced, which creates the potential for substantial variation in timing between participants, both in terms of exposure time to different trial types and in the time elapsed between the start of the FT-IRAP and the imagery tasks. Again, however, there are no reasons to assume systematic differences in timing between the two groups. The alternative would be to introduce timing constraints into the FT-IRAP, but this may prevent participants from responding naturally.

In conclusion, the current study provides preliminary evidence that episodic simulation can enhance implicit expectancies about potential positive future events. Our findings extend previous research into the effect of episodic simulations on explicit expectancies and overcome the limitations inherent in the use of self-report measures. These findings provide a platform for future research using repeated inductions of episodic simulation and inform the development of prospection-focused interventions for depression.

## Author Contributions

**Conceptualization:** Rachel J. Anderson, Kevin J. Riggs, Stephen A. Dewhurst.

**Formal analysis:** J. Helgi Clayton McClure.

**Funding acquisition:** Rachel J. Anderson, Kevin J. Riggs, Stephen A. Dewhurst.

**Investigation:** J. Helgi Clayton McClure, Emma Bishop.

**Methodology:** Rachel J. Anderson, David Howe.

**Project administration:** J. Helgi Clayton McClure.

**Software:** David Howe.

**Supervision:** Rachel J. Anderson.

**Writing – original draft:** Rachel J. Anderson, J. Helgi Clayton McClure, David Howe, Stephen A. Dewhurst.

**Writing – review & editing:** Kevin J. Riggs.

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
