## [Decision Letter · Decision Letter 0]

27 Oct 2023

PONE-D-23-29155The implicit power of positive thinking: The effect of positive episodic simulation on implicit future expectanciesPLOS ONE

Dear Dr. Anderson,

Thank you for submitting your manuscript to PLOS ONE. After careful consideration, we feel that it has merit but does not fully meet PLOS ONE’s publication criteria as it currently stands. Therefore, we invite you to submit a revised version of the manuscript that addresses the points raised during the review process.

 Please submit your revised manuscript by Dec 11 2023 11:59PM. If you will need more time than this to complete your revisions, please reply to this message or contact the journal office at plosone@plos.org. Please include the following items when submitting your revised manuscript:A rebuttal letter that responds to each point raised by the academic editor and reviewer(s). You should upload this letter as a separate file labeled 'Response to Reviewers'.A marked-up copy of your manuscript that highlights changes made to the original version. You should upload this as a separate file labeled 'Revised Manuscript with Track Changes'.An unmarked version of your revised paper without tracked changes. You should upload this as a separate file labeled 'Manuscript'.

We look forward to receiving your revised manuscript.

Kind regards,

Kymberly D. Young, Ph.D.

Academic Editor

PLOS ONE

Journal Requirements:

**Additional Editor Comments:**

Please respond to each comment by the reviewer. Furthermore, while it is mentioned in the discussion that results did not vary as a function of level of depressive symptomatology, there is no explicit section on this in the results. This is strongly needed. 

Reviewers' comments:

Reviewer's Responses to Questions

**Comments to the Author**

1. Is the manuscript technically sound, and do the data support the conclusions?

Reviewer #1: Partly

2. Has the statistical analysis been performed appropriately and rigorously? 

Reviewer #1: No

3. Have the authors made all data underlying the findings in their manuscript fully available?

Reviewer #1: Yes

4. Is the manuscript presented in an intelligible fashion and written in standard English?

Reviewer #1: Yes

5. Review Comments to the Author

Reviewer #1: This article examines the short-term impact of positive episodic simulation training on implicit optimism in a community sample of college students. It aims to address an important gap in the existing literature on episodic specificity inductions and future thinking training, which have primarily relied on explicit measures to index training/induction effects.

There are important methodological, design, and analysis issues in the present study that should be addressed and clarified. In addition, there are a number of assertions that require additional context or support, without which some conclusions appear overstated or unwarranted based on the present findings.

Please see the attached document for specific comments outlined by section.

6. PLOS authors have the option to publish the peer review history of their article (what does this mean?). If published, this will include your full peer review and any attached files.

Reviewer #1: No

---

## [Author Response · Author response to Decision Letter 0]

22 Jan 2024

Response to reviewers/editors comments can be found in an attachment file entitled 'Response to Reviewers'

---

## [Decision Letter · Decision Letter 1]

29 Jan 2024

PONE-D-23-29155R1The implicit power of positive thinking: The effect of positive episodic simulation on implicit future expectanciesPLOS ONE

Dear Dr. Anderson,

Thank you for submitting your manuscript to PLOS ONE. After careful consideration, we feel that it has merit but does not fully meet PLOS ONE’s publication criteria as it currently stands. Therefore, we invite you to submit a revised version of the manuscript that addresses the points raised during the review process.

We look forward to receiving your revised manuscript.

Kind regards,

Kymberly D. Young, Ph.D.

Academic Editor

PLOS ONE

Journal Requirements:

**Additional Editor Comments:**

Please make the requested correction to page 8. 

Reviewers' comments:

Reviewer's Responses to Questions

**Comments to the Author**

1. If the authors have adequately addressed your comments raised in a previous round of review and you feel that this manuscript is now acceptable for publication, you may indicate that here to bypass the “Comments to the Author” section, enter your conflict of interest statement in the “Confidential to Editor” section, and submit your "Accept" recommendation.

Reviewer #1: All comments have been addressed

2. Is the manuscript technically sound, and do the data support the conclusions?

Reviewer #1: Yes

3. Has the statistical analysis been performed appropriately and rigorously? 

Reviewer #1: Yes

4. Have the authors made all data underlying the findings in their manuscript fully available?

Reviewer #1: Yes

5. Is the manuscript presented in an intelligible fashion and written in standard English?

Reviewer #1: Yes

6. Review Comments to the Author

Reviewer #1: The authors have sufficiently addressed my comments and I recommend acceptance of the manuscript, after following small correction is made:

On p. 8 of the Methods section the revised text re: comparing demographic characteristics between groups currently reads “The two groups did differ significantly”; this should be changed to "did not."

7. PLOS authors have the option to publish the peer review history of their article (what does this mean?). If published, this will include your full peer review and any attached files.

Reviewer #1: No

---

## [Author Response · Author response to Decision Letter 1]

29 Jan 2024

We have made the necessary correction on p8. We thank the reviewer for their detailed comments on the first version of the manuscript and picking up on this error in the revision.

---

## [Editor Report · Decision Letter 2]

31 Jan 2024

The implicit power of positive thinking: The effect of positive episodic simulation on implicit future expectancies

PONE-D-23-29155R2

Dear Dr. Anderson,

We’re pleased to inform you that your manuscript has been judged scientifically suitable for publication and will be formally accepted for publication once it meets all outstanding technical requirements.

Kind regards,

Kymberly D. Young, Ph.D.

Academic Editor

PLOS ONE